# Evaluation of Locomotion Complexity in Zebrafish after Exposure to Twenty Antibiotics by Fractal Dimension and Entropy Analysis

**DOI:** 10.3390/antibiotics11081059

**Published:** 2022-08-04

**Authors:** Michael Edbert Suryanto, Chun-Chuen Yang, Gilbert Audira, Ross D. Vasquez, Marri Jmelou M. Roldan, Tzong-Rong Ger, Chung-Der Hsiao

**Affiliations:** 1Department of Bioscience Technology, Chung Yuan Christian University, Chung-Li 320314, Taiwan; 2Department of Chemistry, Chung Yuan Christian University, Chung-Li 320314, Taiwan; 3Department of Physics, Chung Yuan Christian University, Chung-Li 320314, Taiwan; 4Department of Physics, National Central University, Chung-Li 32001, Taiwan; 5Department of Pharmacy, Faculty of Pharmacy, University of Santo Tomas, Manila 1015, Philippines; 6Research Center for the Natural and Applied Sciences, University of Santo Tomas, Manila 1015, Philippines; 7The Graduate School, University of Santo Tomas, Manila 1015, Philippines; 8Faculty of Pharmacy, University of Santo Tomas, Manila 1015, Philippines; 9Department of Biomedical Engineering, Chung Yuan Christian University, Taoyuan 320314, Taiwan; 10Center for Nanotechnology, Chung Yuan Christian University, Chung-Li 320314, Taiwan; 11Research Center for Aquatic Toxicology and Pharmacology, Chung Yuan Christian University, Chung-Li 320314, Taiwan

**Keywords:** antibiotics, animal behavior, fractal dimension, entropy, phenomics

## Abstract

Antibiotics are extensively used in aquaculture to prevent bacterial infection and the spread of diseases. Some antibiotics have a relatively longer half-life in water and may induce some adverse effects on the targeted fish species. This study analyzed the potential adverse effects of antibiotics in zebrafish at the behavioral level by a phenomic approach. We conducted three-dimensional (3D) locomotion tracking for adult zebrafish after acute exposure to twenty different antibiotics at a concentration of 100 ppb for 10 days. Their locomotor complexity was analyzed and compared by fractal dimension and permutation entropy analysis. The dimensionality reduction method was performed by combining the data gathered from behavioral endpoints alteration. Principal component and hierarchical analysis conclude that three antibiotics: amoxicillin, trimethoprim, and tylosin, displayed unique characteristics. The effects of these three antibiotics at lower concentrations (1 and 10 ppb) were observed in a follow-up study. Based on the results, these antibiotics can trigger several behavioral alterations in adult zebrafish, even in low doses. Significant changes in locomotor behavioral activity, such as total distance activity, average speed, rapid movement time, angular velocity, time in top/bottom duration, and meandering movement are highly related to neurological motor impairments, anxiety levels, and stress responses were observed. This study provides evidence based on an in vivo experiment to support the idea that the usage of some antibiotics should be carefully addressed since they can induce a significant effect of behavioral alterations in fish.

## 1. Introduction

The use of antibiotics is widespread throughout the world for the treatment of human and animal diseases. The global use of antibiotics, which includes medical, veterinary, and growth factors in the food business, is estimated to be 100–200 tons per year [1,2]. In animal farming, antibiotics were integrated into the feeds or fed directly via oral administration, pond sprinkle, injection, or dispersed into the water for aquaculture [3,4,5,6]. Aquaculture is becoming a more concentrated sector, and it is one of the most sustainable types of agricultural production for human consumption of protein [7]. Aquaculture has continued to increase in recent decades as a result of intensive procedures and antibiotic use, with Asia contributing to approximately 90% of this expansion. The extensive use of antibiotics can lead to agro-ecosystem contamination that endangers non-target animals in the aquatic environment through the water system. Another major route of antibiotic contamination is the use of feces from animals such as swine, chickens, and humans that serve as a nutritional source [6,8]. After defecation from animals or humans, several antibiotics remained biochemically active that will eventually reach the aquatic environment [9]. The antibiotics in the water environment can be found at concentrations from ng/L up to μg/L. For example, up to 300 ng/L of erythromycin was found in the water surface surrounding animal farms [10]. In municipal sewage, tetracycline was found in the range of 0.7 to 65.2 μg/L [11]. Meanwhile, another antibiotic, ciprofloxacin, can be detected at higher concentrations ranging from 0.7 to 124.5 μg/L in hospital wastewater [12]. Ionized compounds, including antibiotics, have a high affinity for particular target regions, such as proteins or DNA, which might result in undesirable pharmacological effects in aquatic species [13]. The studies on toxicological effects were mainly analyzing the potential of chemicals bioaccumulation and the specific organism sensitivity. Most studies involved the pharmacodynamics, including absorption, distribution, metabolism and excretion of these antibiotics [1]. 

Based on their chemical structure, antibiotics are divided into several classes: aminoglycoside, cephalosporin, fluoroquinolone, glycopeptide, macrolide, sulfonamide, tetracycline, and β-lactam [3]. Some antibiotics are prescribed for aquatic animals, for humans, and some are utilized for both aquaculture and human medicine. Each antibiotic has specific target diseases and species. Septicemia and skin disorders in fish are routinely treated with a fluoroquinolone [14]. Sulfonamide and tetracycline are commonly used for bacterial infection prophylactics and treatments [3]. In salmon, oxytetracycline and tetracycline are commonly used to treat illnesses such as furunculosis and *Vibrio*, while erythromycin is commonly used to treat bacterial kidney disease. Flatfish, jacopever, and yellowtail vibriosis are treated with tribrissen, an antibiotic with numerous active components (5:1 mixture of sulfadiazine and trimethoprim) [15,16]. Meanwhile, in humans, antibiotics such as penicillin have been widely used for treating infectious diseases, immunomodulatory treatments, organ transplantation, and many other medical treatments [17,18]. Regardless of their intended use, several antibiotics were found in aquaculture products and surroundings [19]. As a result of the large-scale and indiscreet use of antibiotics, drug residues have accumulated in the natural environment.

Antibiotics’ negative impacts on aquatic organisms were reported, including effects on survival, growth, reproductive ability, and biochemical markers alterations; hence, antibiotics affect the entire food chain of aquatic organisms. However, the consequences of antibiotic exposure on aquatic animal behavior have rarely been studied [3,20]. The use of behavioral changes to evaluate the sublethal effects of chemicals on non-target aquatic creatures is becoming increasingly popular [21]. In toxicology, the zebrafish (*Danio rerio*) is a vertebrate model that is widely used to study toxic substances based on changes in behavior and neurodevelopment [22,23]. Zebrafish were reported to have around 190 behavioral-related traits [24], both in the wild and in laboratory conditions. This species prefers its conspecifics and forms shoals and dominance hierarchies. They show neophiliac responses to new environments/situations, as well as preferences for depth, color, and darkness. Both sexes have shown aggressive, fear/anxiety-like, and particular breeding (“courting”) behaviors [24,25,26]. Various chemical substances have already been widely tested on zebrafish behavior by using their visual motor response [27]. In recent years, multiple endpoints have been employed for fish behavioral assessment, including swimming speed [28], moved distance [22,27], angular velocity [16], thigmotaxis preference [29], and feeding performance [30]. Changes in these endpoints are linked to external stress, anxiety level, and neurotoxicity. 

The study of zebrafish behavior is increasingly being used in toxicology to track phenotypic changes because of its well-characterized responses [31]. Various behavioral tests seem ideal for evaluating the effects of pollutants on fish populations because their behaviors are highly related to the responses to internal and external changes [32]. Recent findings suggest that zebrafish are more resilient than rodents in terms of behavioral responses [33]. In this study, adult zebrafish were used to investigate the behavioral effects of acute exposure to environmentally appropriate doses of twenty antibiotics chosen to represent widely used antibiotic classes. In this study, we also used fractal dimension and entropy analysis to evaluate the locomotion complexity in zebrafish after exposure to twenty antibiotics. In addition, a follow-up investigation was conducted with lower doses of amoxicillin, trimethoprim, and tylosin based on the result of the phenomic analysis. 

## 2. Materials and Methods

### 2.1. Zebrafish Maintenance 

Golden zebrafish (4–6 months old) were purchased from the aquarium store in our local area (Zhongli District, Taoyuan, Taiwan). Despite their ability to acclimatize quickly to the new environment, the fish were kept in the zebrafish facility for one month before the test to eliminate the effects of external factors. The zebrafish were maintained in the aquatic recirculating system at 28 ± 1 °C with pH 7.0–7.5 and 10/14 h dark/light cycle. Throughout the day, the water was continually filtered by ultraviolet (UV) light with a conductivity of 300–1500 µS. Two times a day, the fish were fed with brine shrimp (*Artemia*) or commercial flake foods. The housing procedures were based on the previous publication [34]. The experimental protocols were conducted in accordance with Chung Yuan Christian University’s Institutional Animal Care and Use Committees (IACUCs) (CYCU109001, 2 January 2020).

### 2.2. Antibiotic Preparation and Exposure

Twenty antibiotics of Amikacin, Amoxicillin, Azithromycin, Cefuroxime, Ciprofloxacin, Doxycycline, Erythromycin, Gentamycin, Norfloxacin, Ofloxacin, Oxytetracycline, Penicillin G potassium salt, Streptomycin, Sulfamethoxazole, Sulfamethazine, Sulfapyridine, Tetracycline, Trimethoprim, Tylosin, and Vancomycin that belong from different classes (Table A1 in Appendix A) were purchased from Shanghai Macklin Biochemical Co., Ltd. (Shanghai, China). These antibiotics were selected based on their high define daily doses (DDD) value and sustainability in water (Table A1). Stock solutions at 1000 ppm were prepared with ddH_2_O and serially diluted to working concentration of 100 ppb. The concentration was selected based on maximum antibiotic concentrations detected in the aqueous environments in Taiwan [35]. For both control and antibiotic-treated groups, seven mixed-gender adult golden zebrafish were used and incubated in 3 L fish water tank. The control group did not receive any other treatment. Meanwhile, for antibiotic treatment, an antibiotic with the desired working concentration was immersed in the water. The aeration was continuously supplied to the glass tanks throughout the experiment for 10 days. Fish were fed with fresh hatched *Artemia* or commercial flake foods twice a day. The water in each glass tank was entirely replaced with new aquarium water every two days and for antibiotic-treated groups, the antibiotic solution was replenished until it reached the final concentration. This approach is carried out to keep the tank in clean condition and concentration of antibiotic exposure constant throughout each group. This experiment was conducted in duplicate. Later, further testing of three selected antibiotics (Amoxicillin, Trimethoprim, and Tylosin) were conducted by using lower doses of 1 and 10 ppb following the same protocol.

### 2.3. Three-Dimensional (3D) Locomotion Tracking

The 3D locomotion test was conducted in the morning until afternoon at room temperature (25 ± 1 °C). After the incubation, the tested fish were put in a polypropylene tank (20 cm × 20 cm × 20 cm) with light-emitting diode (LED) light as the background light sources on the bottom of the back of the tank. The top border of the tank is equipped with a mirror tilted at 45° to create 3D image reflection. A Canon EOS 600D digital camera with a 55–250 mm zoom lens (Canon Inc., Tokyo, Japan) was placed within 5 m in front of the tank to record the fish’s swimming behavior. The tested fish were allowed to be acclimated to the new environment for ~5 min prior to the video recording. After acclimation, the zebrafish behaviors were recorded for 5 min. Afterward, the fish trajectories from the recorded video test were processed by using idTracker (http://www.idtracker.es/, accessed on 29 June 2021), an automatic tracking of multiple animal software [36]. Afterward, we calculated some important behavior endpoints (Table A2 in Appendix A) based on these trajectories. All this protocol was conducted based on our previous publication [37].

### 2.4. Locomotion Trajectory Analysis by Using VBA Macro

In this experiment, we tracked a relatively high number of fish (total ~336 fish); it would be labor-intensive if a traditional method was used to compile data. To overcome this limitation, Excel Visual Basic for Applications (VBA) was used to speed up data calculation throughput. The tool automates Excel tasks and applies them to multiple sheets on a large scale [38]. As far as we know, this is the first study to use VBA to analyze trajectory data. The ability of Excel VBA to run multiple calculations simultaneously makes it superior to regular Excel. The purpose of VBA was to enhance and automate applications, giving users the capability to automate repetitive tasks [39]. To initiate the command for the calculation task, we used a simple record macro function to record the data compilation, calculation, formula, and arrangement in the first batch, then saved it as a ‘hotkey’. After that, by typing the hotkey, the whole process can be performed in succeeding batches. By using these VBA macro scripts, we were able to increase efficiency and reduce the occurrence of human errors. The VBA function can be activated in Excel through the ‘Options’ panel, choose ‘Customize Ribbon,’ then tick the ‘Developer’ tabs. We activated ‘Record Macro’ through the ‘Developer’ tabs located in Tab Menu Bar. Data analysis and calculation were performed in Microsoft Office Professional Plus 2016, Excel Version 2108 (Build 14326.20404).

### 2.5. Fractal Dimension and Entropy Analysis 

The correlation dimension was used in this study. The 3D position of x, y, and z within each frame from fish swimming movement and direction were generated from idTracker. Their swimming patterns and responses in spatial complexity within time–space structured property of zebrafish movement were evaluated by fractal dimension and entropy analysis. The mathematic calculation for fractal dimension and entropy was conducted based on our previous publication [40].

### 2.6. Principal Component Analysis (PCA) and Hierarchy Clustering 

Using Microsoft Excel, the values collected from each behavioral endpoint were summarized into a comma-delimited (csv) file. After that, the csv file was imported to ClustVis (https://biit.cs.ut.ee/clustvis/, accessed on 29 June 2021), to conduct PCA and hierarchy clustering analysis [41]. Each behavior endpoint’s average data was calculated in one excel sheet, then stacked on top of each other as one group. This process was carried out for all groups (control and antibiotic treatments), then PCA plot and hierarchical clustering analysis were performed. PCA plot analysis was applied among the treatments to characterize the variations in community composition. Since the dataset did not contain any missing values, no logarithmic transformation was applied to calculate principal components and hierarchical clustering analysis following our previous protocol [32]. 

### 2.7. Statistics

The mean value of each locomotion endpoint for each antibiotic group was statistically compared to the control group using one-way ANOVA followed by Dunnett’s multiple comparisons test [42]. The data are expressed as mean ± standard error of the mean (SEM) and statistical significances are indicated as * *p* < 0.05; ** *p* < 0.01; ** *p* < 0.001; and **** *p* < 0.0001. All of statistical analyses in this study were conducted by using GraphPad Prism software (San Diego, CA, USA, https://www.graphpad.com/, accessed on 11 July 2021).

## 3. Results

### 3.1. Swimming Movement Activity Assessment by 3D Locomotion Test

The zebrafish behavior after antibiotic treatments was evaluated in the 3D novel tank as illustrated in Figure 1A. The multiple behavior endpoints from each antibiotic group were then compared to the control group. First, we assessed their swimming movement. Fish swim by sweeping the tail fin from side to side. The swimming speed is related to the frequency of the tail sweeps. The fish can steer, glide, turn rapidly or break suddenly and stop the movement by freezing a shaped wave on its body and fins. In this study, we classified the swimming movement into three states: freezing, swimming (normal/regular), and rapid movement. The ratio of these three swimming movement states was calculated. This revealed that some antibiotics could alter the swimming movements ratio. Three antibiotics displayed a significant increase in the freezing movement ratio: Amoxicillin (*p* < 0.0001), Trimethoprim (*p* = 0.0030), and Tylosin (*p* < 0.0148), when compared to the control group (Figure 1B). The antibiotic treatment also affected the swimming time ratio in fish as observed in Ofloxacin (*p* = 0.0491), Oxytetracycline (*p* = 0.0075), Sulfamethoxazole (*p* = 0.0080), Sulfapyridine (*p* = 0.0025), Trimethoprim (*p* = 0.0051), Tylosin (*p* = 0.0058), and Vancomycin (*p* = 0.0425), these groups induced a significant increase in regular swimming movement time ratio compared to control (Figure 1C). On the contrary, based on the increased normal swimming movement time, Oxytetracycline (*p* = 0.0144), Sulfamethoxazole (*p* = 0.0115), Sulfapyridine (*p* = 0.0044), Trimethoprim (*p* = 0.0002), and Tylosin (*p* < 0.0001) displayed a significant decrease in the rapid movement time ratio (Figure 1D). The majority of antibiotic treatments displayed a relaxed swimming movement in this animal.

### 3.2. Exploratory Behavior Assessment by 3D Locomotion Test

Next, we observed the fish’s exploratory behavior after being exposed to antibiotics. The observation tank test was divided into three equal size areas (top, middle, and bottom), and the following endpoints were quantified: the duration of time in the top, middle, and bottom area; distance to the center of the tank (thigmotaxis); total distance traveled in the top area; and total entries to the top area. It was shown that some antibiotics could induce changes in fish exploratory behavior by significantly increasing the duration time in the top (Figure 2A), as displayed by Erythromycin (*p* = 0.0056), Gentamycin (*p* = 0.0283), Sulfamethazine (*p* = 0.0042), and Tetracycline (*p* = 0.0068). Meanwhile, Ofloxacin (*p* = 0.0138), Sulfamethoxazole (*p* = 0.0485), and Sulfapyridine (*p* = 0.0461) reduced the time duration in the top. Some antibiotics caused an alteration in the duration of time spent in the middle area (Figure 2B), with Sulfamethoxazole (*p* = 0.0386) displaying a significantly lower time. In contrast, Ciprofloxacin (*p* = 0.0003), Oxytetracycline (*p* = 0.0037), and Tetracycline (*p* = 0.0392) displayed a significantly higher time when compared to the control. Following the time duration in the top, the duration of time in the bottom also became altered oppositely. The antibiotic groups that previously displayed a significant increase in time spent in the top area had a decreased time spent in the bottom area and vice versa. Ciprofloxacin (*p* = 0.0029), Erythromycin (*p* = 0.0058), Gentamycin (*p* = 0.0047), and Tetracycline (*p* = 0.0008) significantly decreased the time percentage in the bottom area (Figure 2C). On the other hand, Ofloxacin (*p* = 0.0196), Sulfamethoxazole (*p* = 0.0067), and Sulfapyridine (*p* = 0.0076) significantly increased the time percentage in the bottom area. The preference to stay in the top area was also displayed by some antibiotics in the other two supporting endpoints: total distance traveled in the top and total entries to the top area (Figure 2E,F). On the other hand, some antibiotic groups: Amoxicillin (*p* = 0.0004), Streptomycin (*p* = 0.0067), and Tetracycline (*p* < 0.0001), displayed unique behavior showing an altered thigmotaxis behavior which significantly reduced the distance to the center of the tank (Figure 2D). Meanwhile, Cefuroxime (*p* = 0.0471) and Sulfamethazine (*p* = 0.0306) significantly increased the thigmotaxis distance compared to the control group.

The distance and average speed of golden zebrafish was also altered by some antibiotics (Figure 3A,B). Amikacin (*p* = 0.0036) was able to increase the total distance activity significantly compared to the control, while Amoxicillin (*p* = 0.0046), Trimethoprim (*p* < 0.0001), and Tylosin (*p* = 0.0004) significantly decreased it. Compared to the total distance activity, the average speed was also altered in the same manner (Figure 2B). The average angular velocity can be calculated by dividing the change in the angular coordinate by the change in time. It points in the direction of the axis of rotation. Sulfamethoxazole (*p* = 0.0015) could increase the average angular velocity, while Trimethoprim (*p* = 0.0099) reduced it (Figure 3C). Another endpoint, meandering movement, was also observed. Amoxicillin (*p* = 0.0299) and Sulfamethoxazole (*p* = 0.0027) displayed a significant increase in meandering movement (Figure 3D). Meandering would be increased due to high anxiety levels and also during erratic movements.

### 3.3. Dimensional Reduction Assessment of Behavioral Alterations after Antibiotic Exposure

To better extract meaningful information from the 3D locomotion test, fractal dimension (FD) and entropy value were conducted to reduce data complexity. As visualized in Figure A1 (Appendix A), interpreting the behavioral data just based on the swimming trajectories is a challenging task. Thus, we applied the fractal systems which have a self-similarity characteristic that can be quantified as an indicator to measure the number of basic blocks to form a pattern. The FD value from the Amoxicillin- (*p* = 0.0055), Penicillin- (*p* = 0.0121), and Tylosin (*p* = 0.0044)-treated groups are significantly reduced compared to the control group (Figure 4A). In complex biological systems, entropy was also applied as a nonlinear measurement and has aided in understanding the nonlinearity nature of the problem. In this study, the entropy value of zebrafish movement was altered by exposure to some antibiotics. For example, Amikacin (*p* = 0.0489) significantly increased the entropy value, while Oxytetracycline (*p* = 0.0498) significantly decreased it (Figure 4B). In this case, a high entropy value means that the randomness of the motion is high when the movements are spread out, and a low entropy value means the data are nearly concentrated when movements are contained in a particular or specific area.

In order to clarify and have a better observation of behavioral alterations caused by antibiotics, the complex behavior data were then summarized with a dimensionality reduction approach. PCA and hierarchical clustering analysis were applied in this study. Based on PCA results, three antibiotics from three different classes of antibiotics were distinctly separated from the control group: Amoxicillin from the β-lactam class, Trimethoprim from the sulfonamides class, and Tylosin from the macrolides class (Figure 5A). As is displayed in the PCA plot, they are “close together” (not showing much difference), while distinguishing themselves from other compounds. Afterward, hierarchical clustering was implemented to investigate further the components governing behavioral activity during antibiotic treatment. Similar to the PCA result, the hierarchical clustering analysis also clustered and separated those three antibiotics from the other groups, which are positioned in the last three columns (Figure 5B). Hierarchical clustering analysis provided a heatmap dendrogram that gives an overview of similarities and dissimilarities between samples by using the color gradient of the largest (red color) and smallest (blue color) values in the matrix. Supporting each other, the PCA and heatmap results showed those three antibiotics have unique behavioral endpoint patterns that distinguished them from the others. This grouping is plausible since it is matched with the behavioral test results. These three antibiotics identically displayed different expression patterns for behavior activities including up-regulated meandering movement, freezing movement activity, and entropy value. However, they down-regulated the total distance, average speed, and rapid movement activity.

### 3.4. Data Validation for Three Selected Antibiotics of Amoxicillin, Trimethoprim and Tylosin

Based on PCA and heatmap clustering results, three antibiotics, namely Amoxicillin (AMX), Trimethoprim (TMP), and Tylosin (TYL), were selected to conduct further studies due to their unique feature. Zebrafish were exposed to a low dose of either 1 or 10 ppb levels to determine whether such a trace amount of antibiotics could still induce locomotion and exploratory behavior changes. From the total distance activity and average speed, only AMX 1 ppb caused a significantly increased value (*p* = 0.0004 and *p* = 0.0003, respectively) compared to the control group (Figure 6A,B). Meanwhile, TYL at 1 ppb (*p* = 0.0021) and 10 ppb (*p* = 0.0429) induced a significant increase in average angular velocity when compared to the control (Figure 6C). Changes in movement patterns were also observed in the antibiotic-treated groups, characterized by the alteration in meandering movement. The AMX at 1 ppb (*p* < 0.0001), AMX at 10 ppb (*p* = 0.0034) and TMP at 10 ppb (*p* = 0.0416) significantly decreased the meandering degree compared to the control (Figure 6D). Regarding the swimming movement behavior, most of the low-dose antibiotic groups significantly reduced the freezing time movement (Figure A2A in Appendix A). At the same time, only AMX at 1 ppb significantly increased the rapid movement time ratio (Figure A2C in Appendix A).

Next, the exploratory behavior of animals exposed to low-dose antibiotics of AMX, TMP, and TYL was analyzed. Results showed that animals exposed to TMP at 1 ppb (*p* = 0.0082) and 10 ppb (*p* = 0.0005) displayed a significant decrease in the duration of time in the top (Figure 7A). TMP at 10 ppb further significantly increased (*p* = 0.0005) the duration of time in the bottom compared to the control (Figure 7C). These data suggest that animals exposed to TMP have an increase in their anxiety levels. However, another antibiotic group, AMX at 1 ppb, significantly increased the total distance traveled in the top (*p* = 0.0259) and the total entries to the top area (*p* = 0.0108) (Figure 7E,F). Contrary to TMP, the AMX in a low dose reduced the anxiety levels in the animals.

## 4. Discussion

### 4.1. Several Antibiotics Were Not Induced Toxicity Behavior Alterations 

The most important finding of this study is that we demonstrated that the natural behaviors of adult zebrafish are affected by antibiotic drugs, even in low dosages at a ppb level. After antibiotic exposure, changes in the swimming activity, movement patterns, exploratory behavior, and complexity of locomotion behavior were observed. In the screening section with a concentration of 100 ppb, most of the antibiotics that were tested could at least affect one of the locomotor behavior endpoints, except Azithromycin (AZT), Cefuroxime (CXM), Doxycycline (DOX), and Norfloxacin (NOR), which showed no alterations in all endpoints tested in this study. In a previous study, Shiogiri et al. (2017) reported that AZT exposure (5 × 10^4^ and 10 × 10^6^ ppb) to tilapia caused minor damage to the liver, gills, and no lesions in the kidneys were observed [43]. A similar result was also demonstrated in Chinook salmon exposed orally to AZT (30 × 10^3^ ppb), AZT did not cause significant histopathological changes in the gills, kidneys, liver, and heart [44]. However, there are no related studies regarding the effect of AZT exposure on fish behavior. CXM is also considered less toxic as no report showed severe effects occurred after its administration. Only very high doses of CXM (5 × 10^6^ ppb) can induce ocular toxicity with anterior and posterior inflammation in patients [45]. Kamal et al. (2019) reported no threatening impact of CXM administration on mothers undergoing cesarean delivery, as it also does not significantly affect the gut microbiota composition [46]. Some studies reported DOX-induced toxic effects on zebrafish at higher concentrations (20 × 10^3^ and 40 × 10^3^ ppb) with significant changes in glycogen and protein content in the muscles and gills [47]. It is also reported that DOX shows anticonvulsant effects and has toxic effects with motor impairments, respiratory problems, and even death in adult mice [48]. However, in this study, the absence of alterations from the DOX group might be due to the lower concentration used. No study reported toxic effects and behavioral alteration in adult zebrafish due to NOR exposure. However, Liang et al. (2020) found that NOR (25 × 10^3^ ppb) induced oxidative stress and caused developmental abnormalities in the early life stage of zebrafish [49]. Meanwhile, in the lower concentrations (2–5 × 10^3^ ppb), no malformation, disruption in body length, and significant changes in hatching and mortality rate were observed.

### 4.2. Most of the Antibiotic-Induced Behavioral Changes in Locomotor Activity as Their Toxicity Was Also Reported in Previous Studies

The rest of other antibiotics that caused significant changes in adult zebrafish behavior were also found to induce alterations and toxicity in previous studies. It was reported that the representative group of antibiotics, such as aminoglycosides, β-lactams, glycopeptides, fluoroquinolones, macrolides, sulfonamides, and tetracyclines cause negative impacts on aquatic organisms, including freshwater algae, zooplankton, and fish [23]. For example, the exposure of gentamycin (aminoglycosides) (2388–9552 ppb) for a group of zebrafish causes cardiotoxicity, ototoxicity, and nephrotoxicity and also affects the fish locomotor behavior [50,51]. The residual of β-lactam antibiotics displayed highly toxic effects on *Vibrio fischeri* and *Daphnia magna* with high a mortality rate [52]. One β-lactam antibiotic, Penicillin (11.79 × 10^3^–1179 × 10^3^ ppb) was found to alter swimming behavior and physiological parameters of *D. magna* [53]. Vancomycin (glycopeptide) showed low toxicity (EC_50_ > 600 × 10^3^ ppb), which was displayed by the growth inhibition of the green alga *P. subcapitata* [54]. Exposure of erythromycin (macrolides) with acute (10–10 × 10^3^ ppb) and chronic (0.05–0.8 ppb) concentration induced histological alterations in rainbow trout gills and liver [55]. The swimming speed of medaka was reduced after 96 h of treatment with erythromycin (2 to 2 × 10^3^ ppb) [56]. After prolonged exposure to 25–200 × 10^6^ ppb of sulfamethoxazole (sulfonamides), several alterations in the biochemical parameters and hematological system of *Cyprinus carpio* were also reported [57]. In a previous study, the acute toxicity of tetracycline was studied in the freshwater fish *Gambusia holbrooki*, which showed histological changes in the gills as well as enzymatic activity, showing that this chemical might have a pro-oxidative impact [58]. Furthermore, Keerthisinghe et al. (2020) showed that adult zebrafish treated with chronic tetracycline exhibit impaired hepatic lipid metabolism, which led to weight gain [59]. Antibiotic toxicity has been researched in aquatic creatures before, but the consequences of these pollutants on behavior have not been completely investigated. Here, adult zebrafish were treated to different classes of individual antibiotics at relatively low concentrations (ppb level) and some alterations in the zebrafish locomotion, exploratory, and movement patterns were found. 

### 4.3. Fractal Dimension and Entropy Value Served as Algorithms to Identify the Differences in Fish Responses Exposed to Antibiotic

Fish have become a popular and simple animal model for toxicity assessment. The potential toxicological effects of different antibiotics, which focus mainly on the early stage of zebrafish, were reviewed well in previous reports [23]. In this study, we further provide a novel contribution to systematically analyzing the antibiotic toxicity for adult zebrafish at the behavioral level for the first time. Even at relatively low concentrations, the major finding is that antibiotics can still trigger behavioral alterations in zebrafish. Another important contribution of this study is that we offer deeper behavioral analysis by evaluating the movement complexity of zebrafish with two important indexes of fractal dimension (FD) and entropy value. Euclidean space is intuition and fundamental space geometry. In Euclidean space, if a cube, square, and line double their length at each edge this will cause them to be eight, four, and two times their original size. In this way, scientists can define any dimension, including a non-integer d value, the so-called fractal dimension (FD). This idea also applies to the phase space. For example, a moving object contains infinity points in an n-dimensional space. Each point is written as {xi1, xi2, xi3...xin), where “i” represents the ith space vector and n is the dimension of the space. Although the object moves in the n-dimensional space and shows infinite points, the trajectory can still be described in a simple FD. According to Mandelbrot (1967), the FD is a single-valued measure of complexity obtained from the focal fish’s 3D movement and swimming coordinates [60]. Lower fractal dimension values were demonstrated to predict poor wellbeing, and identified a more repetitive and less complicated swimming pattern, whereas higher fractal dimension values identified a more complex swimming pattern [61]. 

FD measures the space-filling capacity; while entropies describe the uncertainty and degree of disorder. The notion of permutation entropy, which examines the data sequence, was initially presented by Bandt and Pompe [62]. As a result, the entropies produced are substantially associated with the data point’s next closest neighbor which translates high-dimensional to low-dimensional vectors. These fundamental principles lowered computing complexity and improved information interpretation. Bandt’s basic assumptions are applied to the study of zebrafish behaviors. In practice, we recorded the zebrafish positions in the (x, y, z) 3D space every step of 1/50 s. Obviously, the decision of the next move of the zebrafish correlated to the previous move. We only need to consider the motion type of zebrafish—go ahead or turn. Then, translate this concept to the mathematics language if the angle between the previous and current displacement vector is greater than 90° or not. Here, we need to compare n consecutive data points in the time series to obtain the permutation entropy. For example, if there were 1001 data points and 1000 results, the zebrafish’s decisions to go ahead and to turn are 300 and 700 times, respectively. Then, the permutation entropy is
(1)$$H(2)=-\frac{700}{1000}\log_{2}\frac{700}{1000}-\frac{300}{1000}\log_{2}\frac{300}{1000}≅0.88$$

If the zebrafish always go straight, the *H*(2) is equal to 0. Zero implies that the behavior of zebrafish is not random and totally predictable (go straight). In a similar case, if the fish always changes its direction at every move, then the *H*(2) is 0 too. In contrast, if the zebrafish changes its direction 500 times out of 1000 results, the *H*(2) will reach the maximum value of 1. *H*(2) = 1 means the behavior is almost non-biological, this is the case of coin tossing or the Brownian motion of the particles in the liquid [63]; in this case, the behavior of zebrafish’s motion is unpredictable. For this reason, we can connect the permutation entropy to the swimming behaviors and replace the word “permutation” with “meandering” to emphasize the biological meaning. Typically, zebrafish are more active in the daytime and more stationery at night [64,65]. External stimulus or drugs can also affect their habits. Therefore, the meandering entropy will be an excellent mathematic tool for measuring the effect. The FD and entropy are highly related to the intensity of pain and stress. A previous study demonstrated that the zebrafish given a fin clip has a decreased FD and complexity of movement and increased pain intensity [66]. Another study also showed that entropy and FD were significantly decreased after exposure to chemical stress from 2.5 × 10^3^ ppb of formaldehyde [67]. In this study, some antibiotics (100 ppb) significantly reduced the FD and entropy value. This observation indicates that only a few antibiotics can reduce the complex responses which reflect the consequences of stress and painful treatment [66]. Meanwhile, for the 1 and 10 ppb antibiotic-treated groups, no significant differences in FD and entropy value were observed compared to the control, indicating that these antibiotics in low concentrations are considered less harmful (Figure A3).

### 4.4. Amoxicillin, Trimethoprim, and Tylosin Caused Contrast Behavioral Effects Compared to Other Antibiotics

Based on the PCA and heatmap results, three antibiotics: Amoxicillin (AMX), Trimethoprim (TMP), and Tylosin (TYL), displayed unique behavioral effects on zebrafish compared to the other antibiotics tested in this study. Thus, we further evaluated the effects of lower concentrations of these antibiotics at 1 or 10 ppb on fish behavior. Surprisingly, even at lower concentrations, these antibiotics can still trigger behavioral changes in the zebrafish. AMX significantly increased the total distance activity, average speed, and rapid movement time but significantly reduced the meandering movement. This behavior indicates low anxiety but the stress is still palpable due to the chemical exposure. Meanwhile, treatment with TMP caused a significant decrease in the duration time in the top, indicating a high anxiety level [68]. TYL has significantly increased the average angular velocity. High angular velocity involves bending of the entire body associated with escaping and startling responses due to stress [69]. Compared to previous reports on these three antibiotics (Table 1), our study focuses on the behavior approach to evaluate their toxicological effects. In this study, we detected and documented the effects of very low concentrations of antibiotics on animal behavior. When compared to reproduction or fatality endpoint methods, aquatic toxicity testing utilizing behavioral methodologies in fish is quick and sensitive, and it provides a connection between physiological and ecological evaluation [70]. Collectively, this study further supports the utility of zebrafish as an animal model for studying the effects of antibiotics on animal behavior and it also confirms the capacity of video-tracking technology in evaluating the locomotion complexity and fish behavior as essential parameters in ecotoxicological study. In addition, large increases of antibiotic consumption have been reported in recent years. According to a previous report, it is estimated that antibiotic usage increased by 72% across the Europe and Asia regions with AMX being the most consumed broad-spectrum antibiotic [71] and it turns out in this study that this most-used antibiotic belongs to the group that displayed unique alteration effects and triggered behavioral changes, even at lower concentrations. Through this study, we provide evidence of the potentially dangerous effects of the widely used antibiotics and also illustrate the necessity to control antibiotic usage to protect the environment.

## 5. Conclusions

It is important to consider several factors when using drugs (e.g., antibiotics) for the medical treatment of disease, such as their impact on environmental sustainability, animal safety, and also human safety. For this reason, research is required to investigate the environmental and health risks of antibiotic usage in aquaculture. This revealed that most of the commonly used antibiotics could cause behavioral changes in adult zebrafish. In addition, even at a low concentration of 1 ppb of AMX, TMP, and TYL, the behavior alterations were still observed. This result indicates that behavior is an ultra-sensitive marker to evaluate the potential toxicity of drugs, such as antibiotics. However, the effects caused by these antibiotics seem to be still tolerable for the fish since there are no significant changes in pain and stress indicators: both FD and entropy value. The use of antibiotic drugs should be investigated further on a population and an ecological level, especially in aquaculture, where bioaccumulation is possible. The processes behind these behavioral changes, which may be connected to the variety of gut bacteria or brain neurohormone alterations, should also be further explored.

## Figures and Tables

**Figure 1 antibiotics-11-01059-f001:**
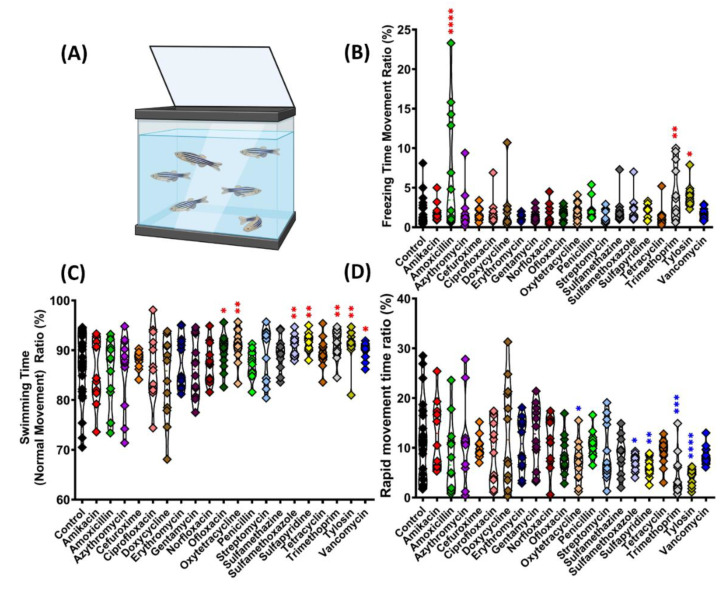
Comparison of swimming movement activity of golden zebrafish after 10 days exposure with different antibiotics in 3D locomotion test. (**A**) Outlooking of 3D locomotion testing tank; (**B**) freezing movement time ratio; (**C**) swimming time ratio; and (**D**) rapid movement time ratio. The data were analyzed by ordinary one-way ANOVA followed by Dunnett’s multiple comparison test (*n* control: 24, *n* for each antibiotic group: 12). The significances were indicated by * *p* < 0.05; ** *p* < 0.01; *** *p* < 0.001; **** *p* < 0.0001 (red asterisk indicates a higher mean value, while blue asterisk indicates a lower mean value than the control).

**Figure 2 antibiotics-11-01059-f002:**
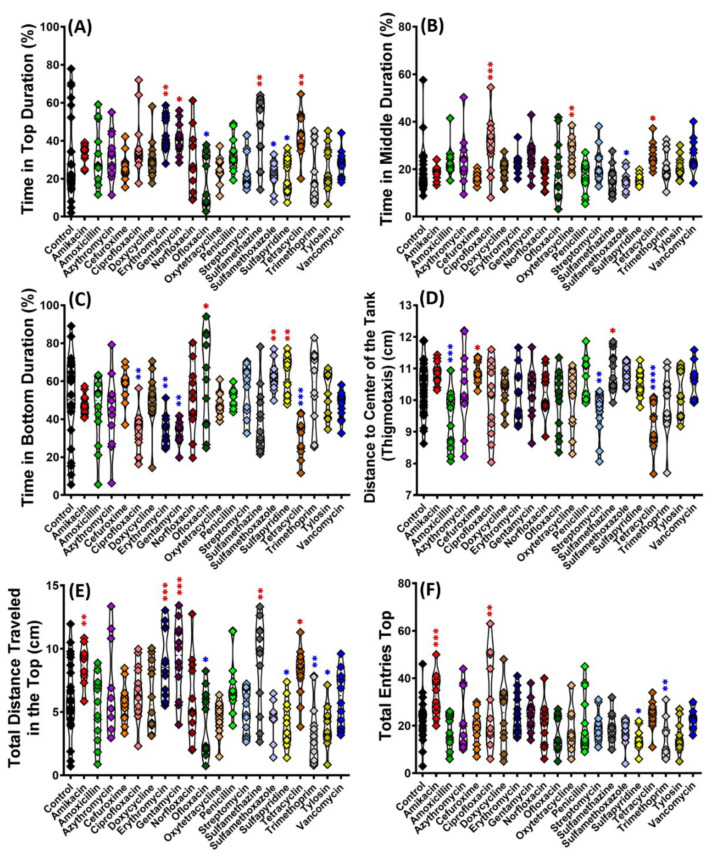
Comparison of exploratory behavior of golden zebrafish after 10 days exposure with different antibiotics in 3D locomotion test. (**A**) Time in top area, (**B**) time in middle area, (**C**) time in bottom area, (**D**) distance to the center of the tank, (**E**) total distance traveled in the top area, and (**F**) total entries in the top area. The data were analyzed by ordinary one-way ANOVA followed by multiple comparison Fisher’s LSD test (*n* control: 24, *n* for each antibiotic group: 12). The significances were indicated by * *p* < 0.05; ** *p* < 0.01; *** *p* < 0.001; **** *p* < 0.0001 (red asterisk indicates a higher mean value, while blue asterisk indicates a lower mean value than the control).

**Figure 3 antibiotics-11-01059-f003:**
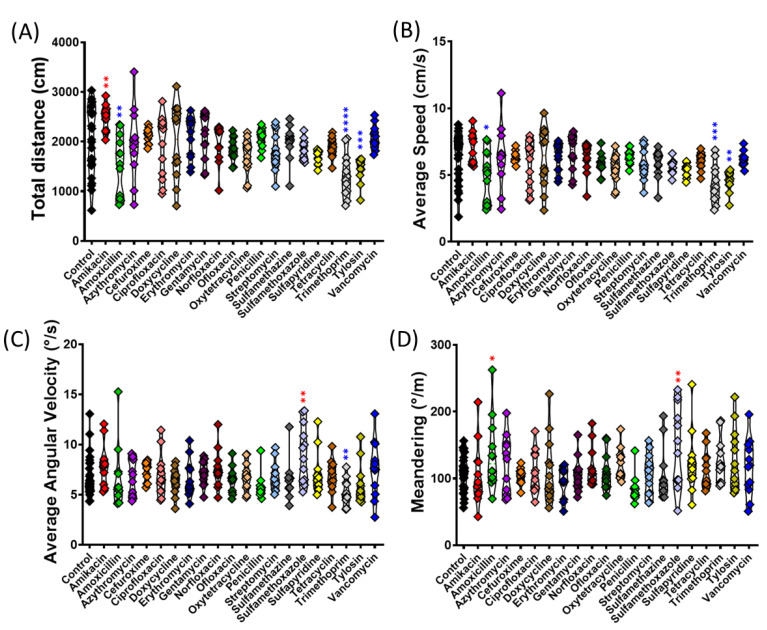
Comparison of locomotor activity endpoints of golden zebrafish after 10 days exposure with different antibiotics in 3D locomotion test. (**A**) Total distance; (**B**) average speed; (**C**) average angular velocity; and (**D**) meandering. The data were analyzed by ordinary one-way ANOVA followed by multiple comparison Fisher’s LSD test (*n* control: 24, *n* for each antibiotic group: 12). The significances were indicated by * *p* < 0.05; ** *p* < 0.01; *** *p* < 0.001; **** *p* < 0.0001 (red asterisk indicates a higher mean value, while blue asterisk indicates a lower mean value than control).

**Figure 4 antibiotics-11-01059-f004:**
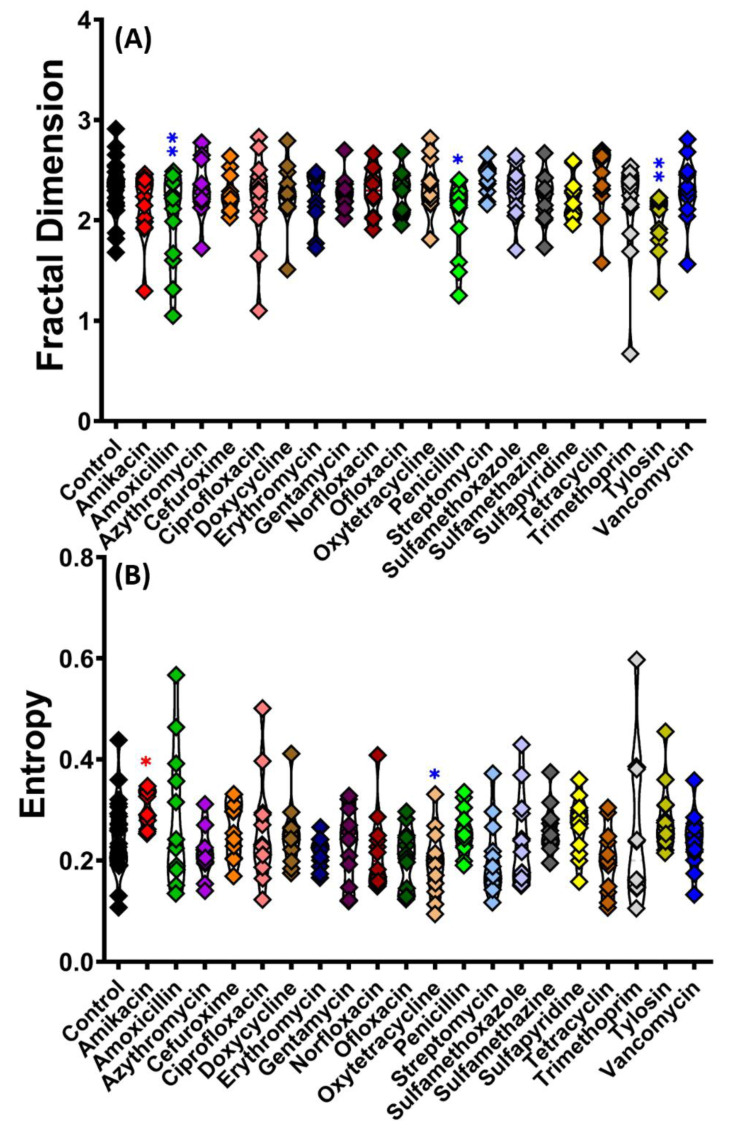
Comparison of (**A**) fractal dimension (FD) and (**B**) entropy value in golden zebrafish locomotion after 10 days exposure with different antibiotics in 3D locomotion test. The data were analyzed by ordinary one-way ANOVA followed by multiple comparison Fisher’s LSD test (*n* control: 24, *n* for each antibiotic group: 12). The significances were indicated by * *p* < 0.05; ** *p* < 0.01 (red asterisk indicates a higher mean value, while blue asterisk indicates a lower mean value than control).

**Figure 5 antibiotics-11-01059-f005:**
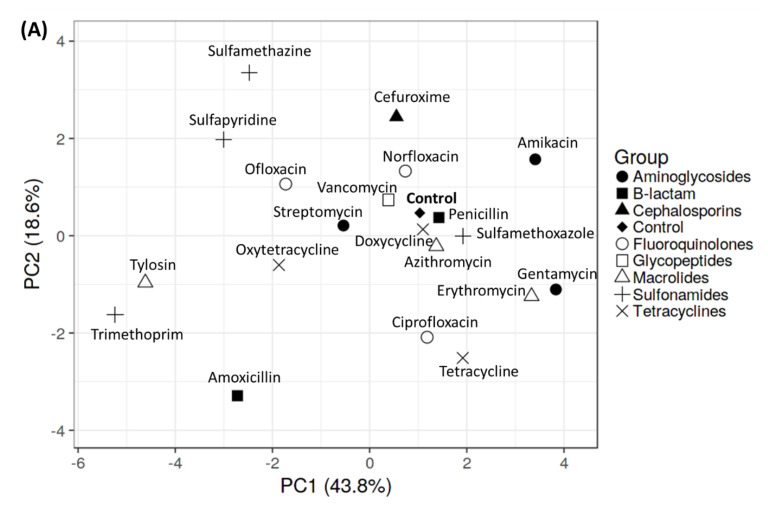
Principal component (PCA) and hierarchical clustering analysis based on multiple behavior endpoints in golden zebrafish after 10 days exposure to different antibiotics. (**A**) PCA projects all behavior endpoints to explore the relationships in data among the antibiotic-treated/non-treated group. (**B**) Hierarchical clustering analysis of different behavior activity post-antibiotic treatment groups. Red indicates higher score of behavior endpoint, whereas blue indicates lower score of behavior endpoint.

**Figure 6 antibiotics-11-01059-f006:**
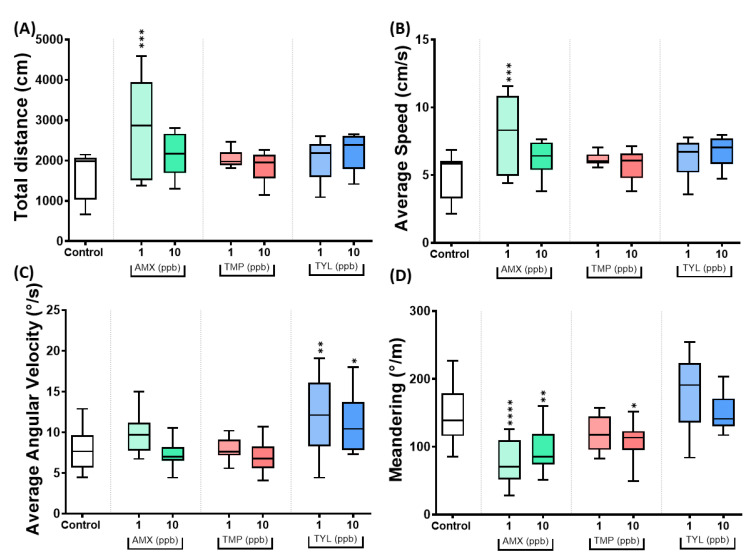
The locomotor activity endpoints of control and antibiotic-treated golden zebrafish after 10 days exposure. (**A**) Total distance traveled (cm); (**B**) average speed (cm/s); (**C**) average angular velocity (°/s); and (**D**) meandering movement (°/m). The data are expressed as mean ± S.E.M and were analyzed by ordinary one-way ANOVA followed by Dunnett’s test (*n* control: 12, *n* for antibiotic groups: 12, except *n* for AMX 1 ppb and TMP 10 ppb: 11, and *n* TYL 10 ppb: 10). The significances were indicated by * *p* < 0.05; ** *p* < 0.01, *** *p* < 0.001, and **** *p* < 0.0001 when compared to control group.

**Figure 7 antibiotics-11-01059-f007:**
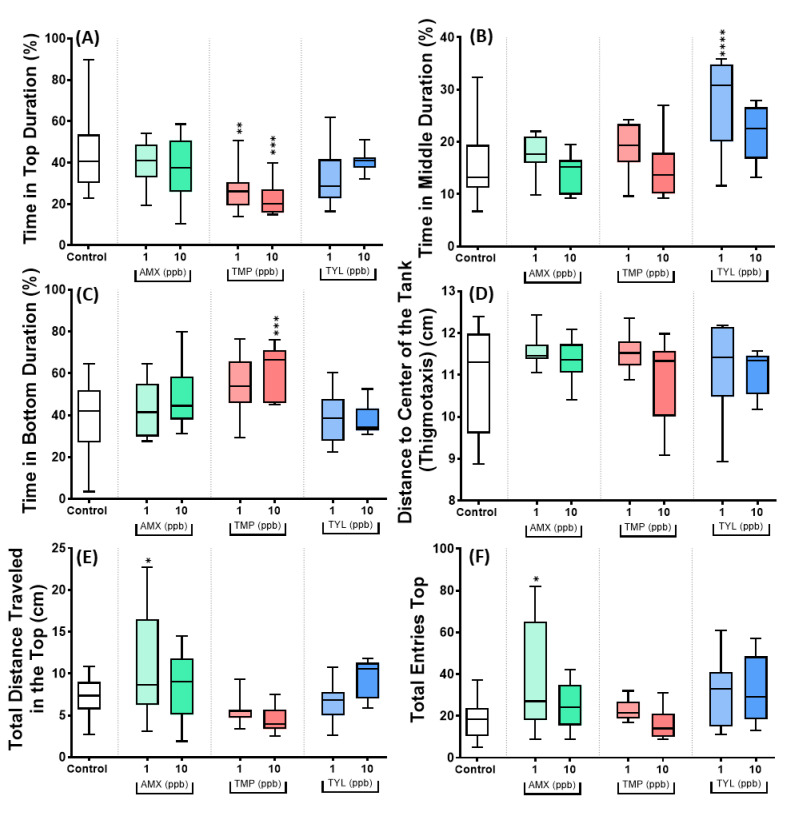
The exploratory behavior of control and antibiotic-treated golden zebrafish after 10 days of exposure. (**A**) Time in top duration (%); (**B**) time in middle duration (%); (**C**) time in bottom duration (%); (**D**) distance to the center of the tank (thigmotaxis) (cm); (**E**) total distance traveled in the top (cm); and (**F**) total entries top. The data are expressed as mean ± S.E.M and were analyzed by one-way ANOVA followed by Dunnett’s test (*n* control: 12, *n* for antibiotic groups: 12, except *n* for AMX 1 ppb and TMP 10 ppb: 11, and *n* TYL 10 ppb: 10). The significances were indicated by * *p* < 0.05; ** *p* < 0.01, *** *p* < 0.001, and **** *p* < 0.0001 when compared to control group.

**Table 1 antibiotics-11-01059-t001:** Previously reported toxicological effects of amoxicillin, trimethoprim, and tylosin on aquatic organisms.

Antibiotics	Concentrations	Exposure Period	Stages	Effects	Refs.
Amoxicillin	100 ppm	7 days	Adult zebrafish	Locomotor alteration and decreased social interaction Induced oxidative stress in brain tissue	[72]
	0, 75, 128, 221, 380, 654 and 1125 ppm	96 h	Embryos and adult zebrafish	Caused premature hatching Induced oxidative stress	[73]
	70 ppm	7 days	Adult zebrafish	Decreased intestinal microbial diversity Increased stress-associated behaviors	[74]
	10, 11, 13, 15, 25, 40, 70, and 80 ppm	96 h	Fingerlings and adult common carp	Behavioral, physical, and biochemical abnormalities Toxicity in liver	[75]
Trimethoprim	0–100 ppm	48 h	In vitro (rainbow trout gonad-2 cell)	Exhibit cytotoxic and genotoxic effects DNA damage and oxidative stress	[76]
	200 ppb	72 h	Zebrafish embryos	Embryotoxicity with reduced hatching rate, body malformations, and high mortality	[77]
	100 ppm	48 h	*Daphnia magna*	Growth inhibition Toxic effect led to death	[78]
	30–300 ppm	5 days	*Daphnia magna* and green algae	Growth inhibition	[79]
Tylosin	5–400 ppb	48 h	Phytoplankton	Growth inhibition Alteration of size structure and composition	[80]
	3–400 ppb	7 days	Green algae	Growth inhibition DNA damage Impairment of molecular pathways related to photosynthesis	[81]
	0, 0.05, 0.2, 1, 5, 25, 100 ppm	96 h	Zebrafish embryos	Decreased survival rate Induced oxidative stress	[82]
	12.5 and 50 ppm	48 h	Zebrafish embryos	Promoted tachycardia and bradycardia	[83]

## Data Availability

The data presented in this study are available directly from the corresponding authors.

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
