# Peer review of "Evaluation of Locomotion Complexity in Zebrafish after Exposure to Twenty Antibiotics by Fractal Dimension and Entropy Analysis"

_antibiotics, 2022, doi:10.3390/antibiotics11081059_

Round 1
Reviewer 1 Report
This paper is very interesting and looking at an important topic in the effects of antibiotics on aquatic species. I think the writing and methods are novel and interesting but I believe that the paper needs more detail in some areas and the study design is slightly lacking in my opinion. I think this paper needs some more work but should definitely be published after some improvement. I ahve added some critiques below.
INTRODUCTION
"active that will eventually reaching the aquatic environment [10]" should be reach instead of reaching.
Spell out twenty rather than writing 20 throughout the paper.
MATERIALS AND METHODS
How old were the zebrafish exactly? Adult isn't very descriptive.
I'm unclear whether there were control fish or if the other doses were actually used in this study. Other doses are listed here. This is more clear in the results section but I think should be made more clear in this section.
RESULTS
Some of the explanation of the PCA/Clustering probably belongs in the methods section
I would like to know how the doses were picked. If you are only going to use one dose for all the compounds and not use the lower doses for all of them then why that does? Is it relevant in some way? Why not use a range of doses?
I would like more detail on what is meant by the hierarchical clustering separating out different antibiotics because that isn't clear to me. I also don't think that PCA separating three compounds out is necessarily relevant unless you have a scientifically sound explanation to go along with it. Why would it separate those three out? How are they different?
Author Response
Comments and Suggestions for Authors
This paper is very interesting and looking at an important topic in the effects of antibiotics on aquatic species. I think the writing and methods are novel and interesting but I believe that the paper needs more detail in some areas and the study design is slightly lacking in my opinion. I think this paper needs some more work but should definitely be published after some improvement. I ahve added some critiques below.
The authors highly appreciate the reviewer for taking necessary time and effort to review this manuscript.
INTRODUCTION
"active that will eventually reaching the aquatic environment [10]" should be reach instead of reaching.
Spell out twenty rather than writing 20 throughout the paper.
Thank you so much for your corrections. The word “reaching” thus has been changed to “reach” (line 56) and the number “20” has been changed to “twenty” as your suggestion.
MATERIALS AND METHODS
How old were the zebrafish exactly? Adult isn't very descriptive.
The authors thank the reviewer for addressing this matter. The adult zebrafish 4-6 months old were used in this study. Therefore, an additional information regarding the age of zebrafish that used in this study has been added in the Materials and Methods (zebrafish maintenance section, line 116).
I'm unclear whether there were control fish or if the other doses were actually used in this study. Other doses are listed here. This is more clear in the results section but I think should be made more clear in this section.
The authors thank the reviewer for pointing out this matter. We apologize for unclear description regarding the control fish and other lower doses used in this study. Therefore, additional description was added to the section (line 137-149). Now, in this section, we have mentioned the control treatment and the other lower doses in the end of this paragraph. Hopefully, this will clarify this matter. Thank you.
RESULTS
Some of the explanation of the PCA/Clustering probably belongs in the methods section
Thank you for your suggestion. Thus, as your suggestion, the explanation of PCA and hierarchical clustering in the result section have been moved to the methods section (line 192-197).
I would like to know how the doses were picked. If you are only going to use one dose for all the compounds and not use the lower doses for all of them then why that does? Is it relevant in some way? Why not use a range of doses?
Thank you for pointing out this matter. The initial concentration that used in this study is 100 ppb and it was selected based on previous literatures that showed the environmental concentrations of antibiotics that were detected in several hospital and municipal wastewater to be in µg/L range (Robinson et al., 2005). Furthermore, other report displayed the maximum antibiotic concentrations detected in the aqueous environments in Taiwan were approximately 0.1 mg/L (Lin et al., 2008). This additional information has been added to line 135-137. The single dosage used in this study was set at 100 µg/L (ppb) as the high estimated environmental concentration (EEC) value to conservatively represent the worst-case scenario of direct exposure of this pharmaceutical contamination. For this reason, we initially used this single dose (100 ppb) for screening the twenty antibiotics based on high EEC value and it turned out that there were three antibiotics which displayed the most distinctive locomotion behavior alteration. So, further testing was conducted to evaluate those three selected antibiotics with lower doses: 1/10 and 1/100 from the initial dose.
Robinson, April A., Jason B. Belden, and Michael J. Lydy. "Toxicity of fluoroquinolone antibiotics to aquatic organisms." Environmental Toxicology and Chemistry: An International Journal 24.2 (2005): 423-430.
Lin, Angela Yu-Chen, Tsung-Hsien Yu, and Cheng-Fang Lin. "Pharmaceutical contamination in residential, industrial, and agricultural waste streams: risk to aqueous environments in Taiwan." Chemosphere 74.1 (2008): 131-141.
I would like more detail on what is meant by the hierarchical clustering separating out different antibiotics because that isn't clear to me. I also don't think that PCA separating three compounds out is necessarily relevant unless you have a scientifically sound explanation to go along with it. Why would it separate those three out? How are they different?
The authors appreciated the reviewer’s point. In this study, the term of PCA and hierarchical clustering were applied to evaluate the effect of antibiotics based on behavioral activity in zebrafish. This approach is well known as behavioral phenomics analysis. This phenomic concept has been proposed for studying behavioral phenotypes based on endogenous or exogenous factors. So, instead of measuring the phenome changes in response to genetic mutation or biochemical like the regular phenomics analysis; the behavior phenomic analysis was measured based on behavioral traits (Javer et al., 2018). Behavior phenomic approaches had been applied in previous study to screening psychiatric drug (Kokel & Peterson 2008) and toxicity in zebrafish (Audira et al., 2020), and other animals such as pig (Perez & Steibel 2021) and worm (McDiarmid et al., 2018). In this study, the PCA plot and heatmap results separating the antibiotics based on the changes or alterations in the behavior endpoints. The plotting and heatmap were generated from multivariate data which in this case is the average value from each behavioral endpoints tested, then presented as a matrix. PCA visualizes compounds in two-dimensional space in such a way that compounds that are "close together" (i.e., not showing much difference) will appear together on the PCA plot. Meanwhile, heatmap visualizes values in the cells using a color gradient with overview of largest (red) and smallest (blue) values in the matrix (Metsalu & Vilo 2015). Similar to the PCA plot, the heatmap columns that “join together” sooner are more similar to each other than those that join together later. Supporting each other, our PCA and heatmap results showed those three antibiotics have unique behavioral endpoint pattern that distinguished them from others. This grouping is plausible since it is matched with the behavioral test results from those three antibiotics, which identically to upregulated the meandering movement, freezing movement activity, and entropy value, but down regulated the total distance, average speed, and rapid movement activity. Regarding to this matter, additional explanation was added in this revised manuscript to improve the description of PCA and heatmap results (line 324-334).
Javer, Avelino, Lidia Ripoll-Sánchez, and André EX Brown. "Powerful and interpretable behavioural features for quantitative phenotyping of Caenorhabditis elegans." Philosophical Transactions of the Royal Society B: Biological Sciences 373.1758 (2018): 20170375.
Kokel, D., & Peterson, R. T. (2008). Chemobehavioural phenomics and behaviour-based psychiatric drug discovery in the zebrafish. Briefings in Functional Genomics and Proteomics, 7(6), 483-490.
McDiarmid, Troy A., Alex J. Yu, and Catharine H. Rankin. "Beyond the response—High throughput behavioral analyses to link genome to phenome in Caenorhabditis elegans." Genes, Brain and Behavior 17.3 (2018): e12437.
Audira, G., Siregar, P., Chen, J. R., Lai, Y. H., Huang, J. C., & Hsiao, C. D. (2020). Systematical exploration of the common solvent toxicity at whole organism level by behavioral phenomics in adult zebrafish. Environmental Pollution, 266, 115239.
Pérez-Enciso, Miguel, and Juan P. Steibel. "Phenomes: the current frontier in animal breeding." Genetics Selection Evolution 53.1 (2021): 1-10.
McDiarmid, Troy A., Alex J. Yu, and Catharine H. Rankin. "Beyond the response—High throughput behavioral analyses to link genome to phenome in Caenorhabditis elegans." Genes, Brain and Behavior 17.3 (2018): e12437.
Metsalu, Tauno, and Jaak Vilo. "ClustVis: a web tool for visualizing clustering of multivariate data using Principal Component Analysis and heatmap." Nucleic acids research 43.W1 (2015): W566-W570.
Reviewer 2 Report
Manuscript ID: antibiotics-1828911
Title: Evaluation of Locomotion Complexity in Zebrafish After Exposure to 20 Antibiotics by Fractal Dimension and Entropy Analysis
General Comments:
This manuscript describes the effect of 20 different antibiotics on the behavioral responses of adult golden zebrafish. The study is well written and easy to follow and is an appropriate topic to include in the journal. The figures are clear and easy to interpret and the findings in the study are relatively novel. The largest shortcoming of the study is that none of the exposure concentrations were analytically validated and are purely based on nominal values. The authors also fail to provide information on the half life of the products used and that would add a lot of clarity to this study. The authors also include compare their study to a lot of different studies in the literature, however each study has different units and this makes it a real challenge for the reader to contextualize the results. If authors choose to use ppb they should convert the values from other studies to ppb when possible.
Specific Comments:
-Abstract was clear and succinct
-Authors should include line numbers on the manuscript to improve the reviewers experience (makes it easier to make specific comments).
-After the statement “After defecation from animals or humans, several antibiotics re-mained biochemically active that will eventually reaching the aquatic environment” authors should include some concentrations of antibiotics that have been measured near agricultural sites.
-Why were golden zebrafish selected? Have they ever been genotyped?
-Authors should include a table in the supplemental that describes what class each antibiotic belongs to, the relative purity of the stock, and the table should also outline which antibiotics were used at 100ppb as well as 1 and 10ppb. The table should also include the half life of each of the antibiotics used to provide more info to the reader.
-Were the antibiotic exposure concentrations analytically validated or is all the data based on nominal concentrations?
-What age range was used as ‘adult’ fish?
-Was the camera placed above the tank or to the side?
-Why did the authors only record 5 minutes of behavior? That is a very brief time to capture swim behaviors
-The beginning of the results section is a repeat of the methods and does not add any value to the manuscript. Authors should revise this section.
Figure 6 – Box plots would more appropriately display the data as they allow the reader to see the variability In the behavioral responses within each treat,ent group. Authors should also include n values for each treatment in the figure caption.
Figure 7 – same comments as figure 6.
The statement “Several antibiotics are considered safe as no behavior alterations have been observed” is problematic as changes in bevior in a study with a n=12 per treatment can’t be used to assess the ‘safety’ of a drug. This statement should be revised.
How do the exposure concentrations used in the study compare to the application rate included on the label of each of the products? Authors should elaborate on this in the discussion.
Author Response
Comments and Suggestions for Authors
Manuscript ID: antibiotics-1828911
Title: Evaluation of Locomotion Complexity in Zebrafish After Exposure to 20 Antibiotics by Fractal Dimension and Entropy Analysis
General Comments:
This manuscript describes the effect of 20 different antibiotics on the behavioral responses of adult golden zebrafish. The study is well written and easy to follow and is an appropriate topic to include in the journal. The figures are clear and easy to interpret and the findings in the study are relatively novel. The largest shortcoming of the study is that none of the exposure concentrations were analytically validated and are purely based on nominal values. The authors also fail to provide information on the half life of the products used and that would add a lot of clarity to this study. The authors also include compare their study to a lot of different studies in the literature, however each study has different units and this makes it a real challenge for the reader to contextualize the results. If authors choose to use ppb they should convert the values from other studies to ppb when possible.
The authors thank the reviewer for the careful and insightful review of our manuscript. We sincerely appreciate all your valuable comments and suggestions, which helped us in improving the quality of the manuscript. As the reviewer suggested, all antibiotics concentrations from other studies in the literature were converted to ppb value. The half-life of antibiotics have also been included in Table A1.
Specific Comments:
-Abstract was clear and succinct
Thank you for the kind comment.
-Authors should include line numbers on the manuscript to improve the reviewers experience (makes it easier to make specific comments).
Thank you for your suggestions. The line numbers have been added on the manuscript.
-After the statement “After defecation from animals or humans, several antibiotics re-mained biochemically active that will eventually reaching the aquatic environment” authors should include some concentrations of antibiotics that have been measured near agricultural sites.
Thank you for the valuable suggestion. The authors agreed and as the reviewer suggested, some reports about antibiotics’ concentrations that have been detected near agricultural sites were added to manuscript. The antibiotics in water environment can be found at concentrations from ng/L up to μg/L. For example, up to 300 ng/L erythromycin was found in water surface surrounding the animal farms [11]. In municipal sewage, tetracycline was found in the range of 0.7 to 65.2 μg/L [12]. Meanwhile, other antibiotic, ciprofloxacin can be detected at higher concentrations ranging from 0.7 to 124.5 μg/L in hospital wastewater [13].” (line 58-62).
- Christian, T.; Schneider, R.J.; Färber, H.A.; Skutlarek, D.; Meyer, M.T.; Goldbach, H.E. Determination of antibiotic residues in manure, soil, and surface waters. Acta hydrochimica et hydrobiologica 2003, 31, 36-44.
- Liu, H.; Zhang, G.; Liu, C.-Q.; Li, L.; Xiang, M. The occurrence of chloramphenicol and tetracyclines in municipal sewage and the Nanming River, Guiyang City, China. Journal of Environmental Monitoring 2009, 11, 1199-1205.
- Robinson, A.A.; Belden, J.B.; Lydy, M.J. Toxicity of fluoroquinolone antibiotics to aquatic organisms. Environmental Toxicology and Chemistry: An International Journal 2005, 24, 423-430.
-Why were golden zebrafish selected? Have they ever been genotyped?
Thank you for your questions. The golden zebrafish exhibited lightening of the pigmented stripes and golden phenotype. The genotype is characterized by the mutation in slc24a5. The golden zebrafish lacks melanophore pigmentation resulting in a yellow coloration with faint yellow stripes (Lamason et al. 2005). The reason golden zebrafish was selected in this study is due to its availability to purchase in the local pet shop compared to AB strain. Furthermore, previous studies reported that golden zebrafish displayed similar behaviors with AB strain in most of behavioral endpoints such as, aggressiveness, predator avoidance (Audira et al., 2020), social interaction (Barba & Gould 2012), and shoaling (Snekser et al., 2010). In addition, the golden mutant strain also becomes an interesting subject for pigmentation model in the further research. With some reports also showed that the antibiotic was associated with an increased (Karunarathne et al., 2019) or loss pigmentation in zebrafish (Yan et al., 2016), hopefully the use of golden zebrafish in our study might aid other related study in the future.
Lamason, Rebecca L., Manzoor-Ali PK Mohideen, Jason R. Mest, Andrew C. Wong, Heather L. Norton, Michele C. Aros, Michael J. Jurynec et al. "SLC24A5, a putative cation exchanger, affects pigmentation in zebrafish and humans." Science 310, no. 5755 (2005): 1782-1786.
Audira, Gilbert, Petrus Siregar, Stefan-Adrian Strungaru, Jong-Chin Huang, and Chung-Der Hsiao. "Which zebrafish strains are more suitable to perform behavioral studies? A comprehensive comparison by phenomic approach." Biology 9, no. 8 (2020): 200.
Barba‐Escobedo, Priscilla A., and Georgianna G. Gould. "Visual social preferences of lone zebrafish in a novel environment: strain and anxiolytic effects." Genes, Brain and Behavior 11, no. 3 (2012): 366-373.
Snekser, Jennifer L., Nathan Ruhl, Kristoffer Bauer, and Scott P. McRobert. "The influence of sex and phenotype on shoaling decisions in zebrafish." International Journal of Comparative Psychology 23, no. 1 (2010).
Karunarathne, Wisurumuni Arachchilage Hasitha Maduranga, Ilandarage Menu Neelaka Molagoda, Myung Sook Kim, Yung Hyun Choi, Matan Oren, Eui Kyun Park, and Gi-Young Kim. "Flumequine-mediated upregulation of p38 MAPK and JNK results in melanogenesis in B16F10 cells and zebrafish larvae." Biomolecules 9, no. 10 (2019): 596.
Yan, Zhenhua, Guanghua Lu, Qiuxia Ye, and Jianchao Liu. "Long-term effects of antibiotics, norfloxacin, and sulfamethoxazole, in a partial life-cycle study with zebrafish (Danio rerio): effects on growth, development, and reproduction." Environmental Science and Pollution Research 23, no. 18 (2016): 18222-18228.
-Authors should include a table in the supplemental that describes what class each antibiotic belongs to, the relative purity of the stock, and the table should also outline which antibiotics were used at 100ppb as well as 1 and 10ppb. The table should also include the half life of each of the antibiotics used to provide more info to the reader.
Thank you for the valuable suggestion. The authors agreed with the reviewer. To provide more information regarding the antibiotics used in the study, an additional supplemental table that describes the antibiotics class, purity, half-life, and the tested concentrations have been included in manuscript as Table A1.
-Were the antibiotic exposure concentrations analytically validated or is all the data based on nominal concentrations?
Thank you for your question. The antibiotic exposure concentration in this study was selected based on the nominal concentration that represents the maximum antibiotic concentrations detected in the aqueous environments. In Taiwan, it reported that maximum concentration of antibiotic detected in aqueous environments were approximately 0.1 mg/L (100 ppb) (Lin et al., 2008). In previous study, this dose has also been used as the high estimated environmental concentration (EEC) value to conservatively represent the worst-case scenario of direct exposure of the antibiotic contamination (Robinson et al., 2005). For this reason, we initially used this single dose (100 ppb) for screening the twenty antibiotics based on high EEC value and it turned out that there were three antibiotics which displayed the most distinctive locomotion behavior alteration. So, further testing was conducted to evaluate those three selected antibiotics with lower doses: 1/10 and 1/100 from the initial dose.
Lin, Angela Yu-Chen, Tsung-Hsien Yu, and Cheng-Fang Lin. "Pharmaceutical contamination in residential, industrial, and agricultural waste streams: risk to aqueous environments in Taiwan." Chemosphere 74.1 (2008): 131-141.
Robinson, April A., Jason B. Belden, and Michael J. Lydy. "Toxicity of fluoroquinolone antibiotics to aquatic organisms." Environmental Toxicology and Chemistry: An International Journal 24.2 (2005): 423-430.
-What age range was used as ‘adult’ fish?
Thank you for your question. The adult zebrafish with range of 4-6 months old were used in this study. An update has been added to manuscript (line 116).
-Was the camera placed above the tank or to the side?
The authors appreciated the questions from the reviewer. The camera was placed in front of the tank. To clarify this matter, an additional explanation was added to the materials and methods section. “The top border of the tank is equipped with a mirror tilted at 45° to create 3D images reflection. A Canon EOS 600D digital camera with a 55-250 mm zoom lens (Canon Inc., Tokyo, Japan) was placed within 5 meter in front of the tank to record the fish swimming behavior.” (line 154-157).
-Why did the authors only record 5 minutes of behavior? That is a very brief time to capture swim behaviors
Thank you for your question. The duration of 5 minutes for behavior test is conducted based on our previous published method (Audira et al., 2018). Along with many other studies that also applied 5 minutes for the behavior testing time in the novel tank test (Oliveri et al., 2020) (Kalueff et al., 2013) (Pham et al., 2012), which proved that this time is sufficient for behavioral assessment in zebrafish. In the study, the novel tank test was intended to observe the zebrafish behavioral response to the novel environment. Thus, the early minutes is a crucial time to evaluate their movements and responses. In addition, the zebrafish had also been acclimated for ~5 minutes prior to the recording. The longer duration time of recording might cause bias as the zebrafish might be already accustomed to the environment later.
Audira, Gilbert, Bonifasius Putera Sampurna, Stevhen Juniardi, Sung-Tzu Liang, Yu-Heng Lai, and Chung-Der Hsiao. "A simple setup to perform 3D locomotion tracking in zebrafish by using a single camera." Inventions 3, no. 1 (2018): 11.
Oliveri, Anthony N., Megan Knuth, Lilah Glazer, Jordan Bailey, Seth W. Kullman, and Edward D. Levin. "Zebrafish show long-term behavioral impairments resulting from developmental vitamin D deficiency." Physiology & behavior 224 (2020): 113016.
Kalueff, Allan V., Adam Michael Stewart, and Robert Gerlai. "Zebrafish as an emerging model for studying complex brain disorders." Trends in pharmacological sciences 35, no. 2 (2014): 63-75.
Pham, Mimi, Jolia Raymond, Jonathan Hester, Evan Kyzar, Siddharth Gaikwad, Indya Bruce, Caroline Fryar et al. "Assessing social behavior phenotypes in adult zebrafish: Shoaling, social preference, and mirror biting tests." In Zebrafish protocols for neurobehavioral research, pp. 231-246. Humana Press, Totowa, NJ, 2012.
-The beginning of the results section is a repeat of the methods and does not add any value to the manuscript. Authors should revise this section.
Thank you for your constructive feedback. The authors have revised the beginning of results section into “The zebrafish behavior after antibiotic treatments were evaluated in the 3D novel tank as illustrated in Figure 1A. The multiple behavior endpoints from each antibiotic group then were compared to the control group. First, we assessed their swimming movement.” (line 209-211).
Figure 6 – Box plots would more appropriately display the data as they allow the reader to see the variability In the behavioral responses within each treat,ent group. Authors should also include n values for each treatment in the figure caption.
Figure 7 – same comments as figure 6.
The authors appreciate the reviewer’s detailed check. As you suggested, the data in Figure 6 and 7 have been changed to be displayed in box plots to show the variability in the behavioral responses within treatment groups. The n number of each treatment has also been added in the figure caption. In addition, the Figure A2 and A3 have also been changed to box plots to make them uniform.
The statement “Several antibiotics are considered safe as no behavior alterations have been observed” is problematic as changes in bevior in a study with a n=12 per treatment can’t be used to assess the ‘safety’ of a drug. This statement should be revised.
Thank you for your valuable comment and suggestion. The authors agreed with the reviewer; therefore, the statement has been changed to “Several antibiotics were not induced toxicity behavior alterations”. In addition, other statement that indicate the ‘safety’ of a drug was replaced with less toxic or less harmful.
How do the exposure concentrations used in the study compare to the application rate included on the label of each of the products? Authors should elaborate on this in the discussion.
Thank you for your thoughtful and constructive comment. Although we agree that this is an important consideration, the authors believe it is beyond the scope of this manuscript because the exposure concentration used in the study was based on the maximum estimated environmental concentration (EEC) value of antibiotics detected in the aqueous environment. Which obviously this concentration is much lower than the application or consumption rate of antibiotic products. However, the additional data of antibiotics consumption rate based on defined daily dose (DDD) have been added to the supplementary Table A1 to provide more information to the reader. Furthermore, the authors also try to elaborate the antibiotic consumption rate and their relation to our study. Additional sentences were added to the end of discussion section. “In addition, large increases of antibiotics consumption have been reported in recent years. According to the previous report, it estimated antibiotic usage increased by 72% across Europe and Asia-region with AMX as the most consumed broad-spectrum antibiotics [72]. And it turns out in this study, this most used antibiotic belongs to the group that displayed unique alteration effects and trigger behavioral changes, even at the lower concentration. Through this study, we provide evidence of potentially dangerous effects from the widely used antibiotics and also illustrate the necessity to control the antibiotics usage to protect the environment.” (line 529-536). Hopefully, it will be acceptable to you.
Round 2
Reviewer 1 Report
This is an interesting paper which was well done. All of my critiques have been addressed and I think the paper is ready for publication.